# Rational molecular and device design enables organic solar cells approaching 20% efficiency

Jiehao Fu [1,10], Qianguang Yang[2,3,4,10], Peihao Huang[3,4], Sein Chung [5], Kilwon Cho [5], Zhipeng Kan [6], Heng Liu[7], Xinhui Lu [7], Yongwen Lang[1,8], Hanjian Lai[8], Feng He [8], Patrick W. K. Fong [1], Shirong Lu [2] ✉, Yang Yang [9], Zeyun Xiao [3,4] ✉ & Gang Li [1] ✉

For organic solar cells to be competitive, the light-absorbing molecules should simultaneously satisfy multiple key requirements, including weak-absorption charge transfer state, high dielectric constant, suitable surface energy, proper crystallinity, etc. However, the systematic design rule in molecules to achieve the abovementioned goals is rarely studied. In this work, guided by theoretical calculation, we present a rational design of non-fullerene acceptor o-BTP-eC9, with distinct photoelectric properties compared to benchmark BTP-eC9. o-BTP-eC9 based device has uplifted charge transfer state, therefore significantly reducing the energy loss by 41 meV and showing excellent power conversion efficiency of 18.7%. Moreover, the new guest acceptor o-BTP-eC9 has excellent miscibility, crystallinity, and energy level compatibility with BTP-eC9, which enables an efficiency of 19.9% (19.5% certified) in PM6:BTP-C9:o-BTP-eC9 based ternary system with enhanced operational stability.

The rapid development in light-harvesting materials, especially non-fullerene acceptors (NFAs)[1–3], has enabled exciting progress in organic solar cells (OSCs)[4–7]. For the OSCs to be competitive and commercially viable, further improving device performance (power conversion efficiency (PCE) and stability) is urgently needed. Regarding the OSC efficiency, one must improve the open circuit voltage ($V_{OC}$) along with efficient exciton dissociation and charge transport process, i.e., simultaneously suppress $V_{OC}$ loss (the difference between the bandgap and e$V_{OC}$, also known as energy loss, $E_{loss}$) and achieve ideal nano-scale morphology in OSCs. From a physical point of view, minimizing $V_{OC}$ loss requires suppressing the absorption of charge transfer (CT) state

and improving the electroluminescence quantum efficiency (EQE$_{EL}$) of OSC devices[8–10], but the rational design of molecules that can achieve these aims remains a challenge. Besides, the $E_{loss}$ of OSC is not only related to the intrinsic properties of donor (D) and acceptor (A) molecules but also to the nano-morphology of D/A blend, such as the distribution, aggregation and packing of D/A molecules. Therefore, the surface energy and crystallinity of the newly designed molecules should also be considered.

Introducing a proper third component to the benchmark binary blend is a widely used method to further enhance device efficiency, starting from the earlier fullerene era by multiple-donor system to

[1]Department of Electrical and Electronic Engineering, Research Institute for Smart Energy (RISE), Photonic Research Institute (PRI), The Hong Kong Polytechnic University, Hung Hom, Kowloon, Hong Kong 999077, PR China. [2]School of Materials Science and Engineering, Taizhou University, Taizhou 318000, PR China. [3]Thin-Film Solar Cell Technology Research Center, Chongqing Institute of Green and Intelligent Technology, Chongqing School, University of Chinese Academy of Sciences (UCAS Chongqing), Chinese Academy of Sciences, Chongqing 400714, PR China. [4]University of Chinese Academy of Sciences, 100049 Beijing, PR China. [5]Department of Chemical Engineering, Pohang University of Science and Technology, Pohang 37673, South Korea. [6]School of Physical Science and Technology, Guangxi University, Nanning 530004, PR China. [7]Department of Physics, The Chinese University of Hong Kong, Shatin, Hong Kong 999077, PR China. [8]Shenzhen Grubbs Institute and Department of Chemistry, Southern University of Science and Technology, Shenzhen 518055, PR China. [9]Department of Materials Science and Engineering, University of California Los Angeles (UCLA), Los Angeles, CA 90095, USA. [10]These authors contributed equally: Jiehao Fu, Qianguang Yang. ✉e-mail: lushirong@cigit.ac.cn; xiao.z@cigit.ac.cn; gang.w.li@polyu.edu.hk

more choices in NFA era[7,11–18]. Considering the excellent efficiency in state-of-the-art binary NFA OSCs, the third component is naturally better not to disturb the excellent morphology of the host binary blend. In this regard, NFAs with smart halogen substitution positions in end groups have high potential to provide similar molecular orientation and excellent miscibility[19], but adjustable different aggregation and photoelectronic properties. In this sense, the development of such NFAs is likely to achieve the efficiency enhancement goal.

Herein, the molecular design started from the optimization of halogen substitution position in terminal groups. Guided by theoretical calculations, we synthesized o-BTP-eC9, an isomer of BTP-eC9 with different chlorine substitutions on the dichlorinated 1,1-dicyanomethylene-3-indanone (IC-2Cl) end groups[5]. The very similar chemical structures lead to excellent miscibility of these two isomers. Compared to BTP-eC9, o-BTP-eC9 shows shallower LUMO (the lowest unoccupied molecular orbital) level, higher dielectric constant, and weaker crystallinity. When paired with PM6, o-BTP-eC9 based OSC exhibits a comparable PCE of 18.7% with that of BTP-eC9 based OSC (18.9%). And the energy loss in PM6:o-BTP-eC9 device is 41 meV smaller than that in PM6:BTP-eC9 device. Taking advantages of o-BTP-eC9, the good miscibility, the complementary in crystallinity and energetics, we introduced o-BTP-eC9 into the PM6:BTP-eC9 blend to fine tune the nano-scale morphology and the energetics of the host blend. As a result, o-BTP-eC9 induces more favorable phase separation in morphology, more balanced charge transport process and suppressed $V_{OC}$ loss. In PM6:o-BTP-eC9:BTP-eC9 based ternary OSC, a record PCE of 19.9% (19.5% certified by Asymptotic $P_{max}$ Scan) was achieved. The overall rational design paves the way for OSC performance breakthrough.

## Results
### Theoretical calculation and the intrinsic properties of target molecule

The electron-withdrawing ability of IC-2Cl depends on two parts, 2-(4-oxocyclopent-2-en-1-ylidene) malononitrile and dichlorobenzene. As shown in Supplementary Fig. 1, the changes in the position of chlorine atoms have a direct impact on IC-2Cl group's dipole moment, thereby influencing the LUMO electron distribution of NFA molecule. According to the intensity borrowing mechanism, uplifting the LUMO level of NFA is an effective method to suppress the $V_{OC}$ loss of OSC[20,21]. Thus, our design starts from optimizing the dipole moment and uplifting the LUMO level of IC-2Cl group. There are three isomers of dichlorobenzene, namely ortho-dichlorobenzene, meta-dichlorobenzene, and para-dichlorobenzene (Supplementary Fig. 1). Combining with the 2-(4-oxocyclopent-2-en-1-ylidene) malononitrile unit, we can get 6 isomers of the IC-2Cl end groups, denoted as α-o-IC-2Cl, β-o-IC-2Cl, θ-o-IC-2Cl, α-m-IC-2Cl, β-m-IC-2Cl, p-IC-2Cl, respectively (Fig. 1). To select the most promising IC-2Cl group, we calculated the dipole moments, LUMO and HOMO levels of various IC-2Cl groups. From the quantum chemistry calculations, θ-o-IC-2Cl group shows the lowest dipole moment of 3.22 Debye and the highest LUMO level of −3.70 eV, thus becoming our primary target to uplift the LUMO level of NFA.

We then synthesized o-BTP-eC9 following the synthetic routes presented in Supplementary Figs. 2 and 3. And the related chemical structures were carefully characterized with ¹H nuclear magnetic resonance (NMR, Supplementary Figs. 4, 6 and 8), 13C NMR (Supplementary Figs. 5, 7 and 9), gas chromatography-mass spectrometer (GCMS, Supplementary Figs. 10 and 11), single crystal analysis (Supplementary Fig. 12 and Supplementary Data 1 and 2), and matrix-assisted laser desorption/ionization time-of-flight mass spectrometry (MALDI-TOF MS, Supplementary Fig. 13). Figure 2a–c shows the molecular structures of PM6, BTP-eC9 and o-BTP-eC9, respectively. The only difference between the two acceptor materials is the position of chlorine substitution on the IC-2Cl terminal groups. Unlike the opposite distribution of chlorine atoms on the IC-2Cl groups of BTP-eC9, the chlorine atoms in o-BTP-eC9 molecule are placed on the same side as the carbonyl groups. Due to the orientation effect of 2,3-dichlorobenzoyl chloride, the synthetic procedure for θ-o-IC-2Cl is one step less than that for β-o-IC-2Cl, and the over yield of the new end group (54%) is obviously higher than that of its counterpart (36%). Besides the condensation reaction of θ-o-IC-2Cl with dialdehyde gives a yield of 96%, which is also significantly higher than that of the β-o-IC-2Cl (53%).

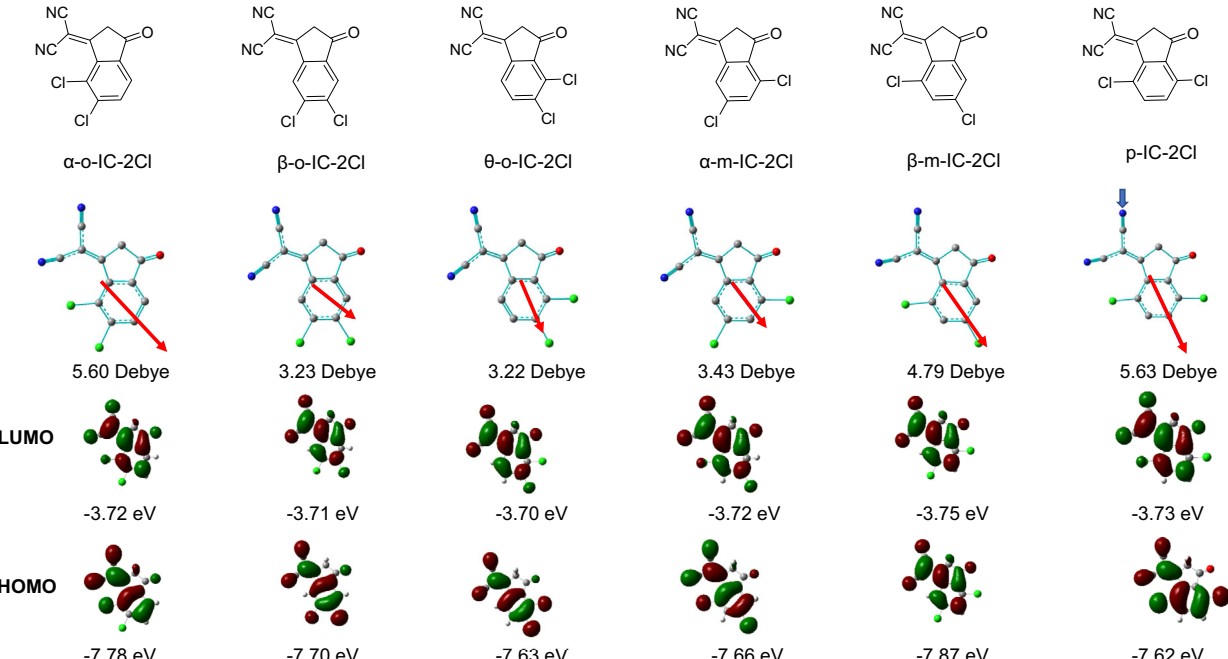

**Fig. 1 | The effect of chlorine substitution position on the property of end group.** Chemical structures of various IC-2Cl groups and the corresponding quantum chemistry calculations of the dipole moments, LUMO and HOMO distributions.

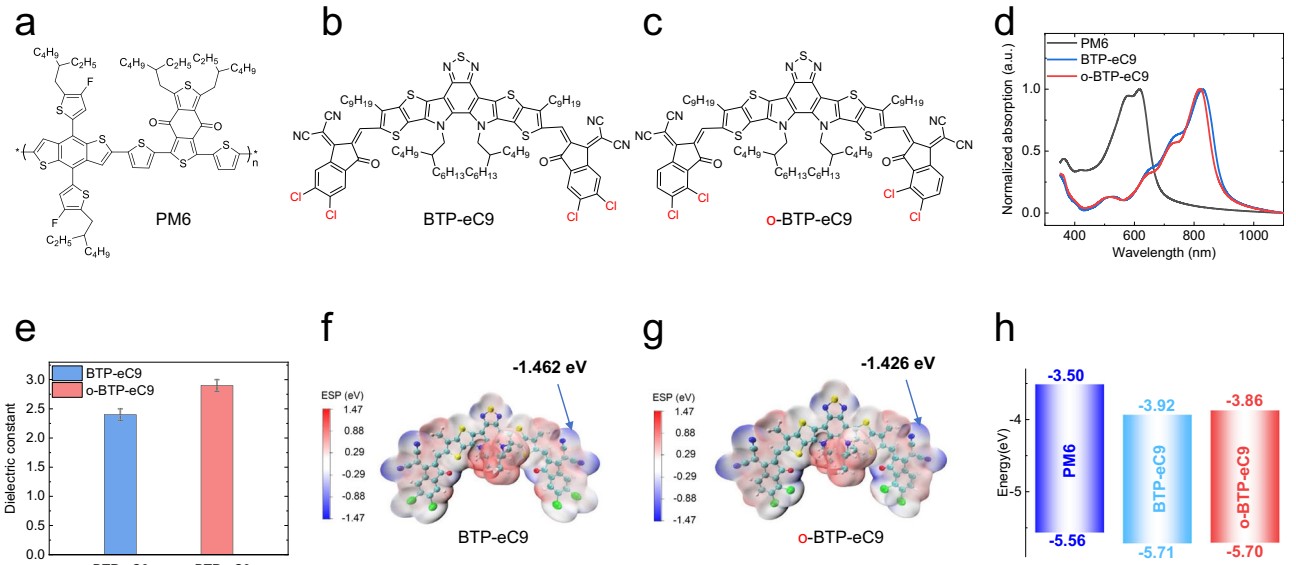

**Fig. 2 | Chemical structures of photoactive materials and the related physical properties.** Chemical structures of PM6 (**a**), BTP-eC9 (**b**), and o-BTP-eC9 (**c**). **d** Normalized UV-vis absorption spectra of PM6, BTP-eC9 and o-BTP-eC9. **e** Comparison of dielectric constant values of BTP-eC9 and o-BTP-eC9. The electrostatic potential surfaces (ESP) of BTP-eC9 (**f**) and o-BTP-eC9 (**g**) calculated from the DFT simulation at the BLYP/6−31G* level (the coordinates of the optimized computational models can be found in Supplementary Data 3). **h** Energy levels of PM6, BTP-eC9 and o-BTP-eC9. Source data are provided as a Source Data file.

**Table 1 | Electrochemical and optical properties of BTP-eC9 and o-BTP-eC9**

| Acceptor | HOMO$^{cv}$[eV] | LOMO$^{cv}$[eV] | $E_g^{cv}$[eV] | $\lambda_{max}^{sol}$[nm] | $\lambda_{max}^{film}$[nm] | $\lambda_{onset}^{film}$[nm] | $E_g^{opt}$[eV] |
|---|---|---|---|---|---|---|---|
| BTP-eC9 | −5.71 | −3.92 | 1.79 | 746.8 | 828.2 | 908 | 1.37 |
| o-BTP-eC9 | −5.70 | −3.86 | 1.84 | 746.8 | 825.6 | 887 | 1.40 |

The ultraviolet−visible (UV−vis) absorption spectra of the two NFAs are presented in Fig. 2d and Supplementary Fig. 14, and the related data are summarized in Table 1. The maximum absorption peaks of BTP-eC9 in solution state ($\lambda_{max}^{sol}$) and thin-film state ($\lambda_{max}^{film}$) are located at 747 nm and 828 nm, respectively, which are consistent with previous report[5]. In comparison, the absorption peak of o-BTP-eC9 solution is unchanged ($\lambda_{max}^{sol}$ at 747 nm) but the absorption peak of o-BTP-eC9 film is slightly blue-shifted ($\lambda_{max}^{film}$ at 826 nm), which is related to the weaker electron-withdrawing ability of the modified IC-2Cl groups (as discussed later). It is obvious that the absorption spectra of both BTP-eC9 and o-BTP-eC9 are red-shifted from solution state to film state, this is a commonly observed phenomenon in the organic photoactive materials and should be ascribed to the enhanced π-π interaction in solid state[22]. The red-shift of o-BTP-eC9 is 79 nm, less than that of BTP-eC9 (81 nm), implying the weaker π-π sacking in o-BTP-eC9 film[22]. Then the dielectric constant ($\varepsilon_r$) values of these two molecules were tested by fabricating capacitors[23] (Supplementary Fig. 15a), and the detailed parameters are presented in Fig. 2e and Supplementary Table 1. BTP-eC9 and o-BTP-eC9 show average $\varepsilon_r$ values of 2.4 and 2.9, respectively, implying the exciton binding energy become smaller in o-BTP-eC9. The higher $\varepsilon_r$ of o-BTP-eC9 should be ascribed to the higher molecular dipole moment[23], as revealed by quantum chemistry calculations at ground state (Supplementary Fig. 15b, c). From the density functional theory (DFT) simulations (Fig. 2f, g and Supplementary Data 3), the minimum negative value of electrostatic potential surface of o-BTP-eC9 (−1.426 eV) is higher than that of BTP-eC9 (−1.462 eV), implying o-BTP-eC9 should show higher LUMO level than BTP-eC9, which is confirmed by electrochemical cyclic voltammetry (CV) measurements. As shown in Supplementary Fig. 16, Fig. 2h and Table 1, o-BTP-eC9 exhibits an obviously higher LUMO energy level and a nearly unchanged highest occupied molecular orbital (HOMO) energy level compared to BTP-eC9.

Correspondingly, the bandgap of o-BTP-eC9 ($E_g^{cv} = 1.84$ eV) is larger than that of BTP-eC9 ($E_g^{cv} = 1.79$ eV), which is in good agreement with the tendency of optical gaps observed from UV-vis absorption spectra (Table 1) and the calculated energy levels (Supplementary Fig. 17).

It is reported that the position of halogen atoms in end groups has a direct impact on molecular stacking[22,24], thus we further investigated the orientation and crystallinity of o-BTP-eC9 and BTP-eC9 by using grazing incidence wide-angle X-ray scattering (GIWAXS) measurements. As revealed in Supplementary Fig. 18 and Supplementary Table 2, both BTP-eC9 film and o-BTP-eC9 film show preferred face-on orientation, with strong π-π stacking (010) peaks at $q \approx 1.68$ Å$^{-1}$ in the out-of-plane (OOP) direction and lamellar (100) peaks at $q \approx 0.43$ Å$^{-1}$ in the in-plane (IP) direction. The corresponding crystallite coherence length (CCL) values of o-BTP-eC9 (CCL$_{010}$ = 15.3 Å and CCL$_{100}$ = 57.1 Å) are smaller than that of BTP-eC9 (CCL$_{010}$ = 18.2 Å and CCL$_{100}$ = 59.5 Å). In general, the changes in chlorine substitution position have little influence on molecular stacking orientation but leads to smaller crystallites, which affords a chance to finely tune the morphology of the active layer by combining o-BTP-eC9 and BTP-eC9 as electron acceptors.

Because the structural difference between BTP-eC9 and o-BTP-eC9 is very small, these two isomers are expected to show excellent miscibility. To verify the hypothesis, we conducted contact angel tests to calculate the surface energy (γ) values of photoactive materials, and to further analyze the miscibility between these materials. As shown in Supplementary Fig. 19, the contact angles of water on PM6, BTP-eC9 and o-BTP-eC9 are 103.5°, 97.4° and 96.1°, respectively, and the contact angles of glycerol on PM6, BTP-eC9 and o-BTP-eC9 are 83.8°, 76.5° and 75.2°, respectively. According to Wu's model[25], the corresponding γ values of PM6, BTP-eC9 and o-BTP-eC9 are calculated as 20.71, 24.94, and 25.57 mNm$^{-1}$, respectively (Supplementary Table 3). The miscibility between material A and B is commonly evaluated by the

equation of $\chi_{A:B} = K(\sqrt{\gamma_A} - \sqrt{\gamma_B})^2$, where $\chi_{A-B}$ is Flory–Huggins interaction parameter, the lower $\chi_{A:B}$ value, the better miscibility between two materials, and $K$ is a positive constant[26,27]. $\chi_{PM6:BTP-eC9}$ and $\chi_{PM6:o-BTP-eC9}$ are calculated as 0.196 $K$ and 0.256 $K$ respectively, while $\chi_{BTP-eC9:o-BTP-eC9}$ is much lower, only 0.0039 $K$, indicating the excellent miscibility between BTP-eC9 and o-BTP-eC9. The significant difference between $\chi_{PM6:o-BTP-eC9}$ and $\chi_{BTP-eC9:o-BTP-eC9}$ implies that the PM6:o-BTP-eC9 blend tends to form microscopic morphology with obvious phase separation, but BTP-eC9:o-BTP-eC9 blend is possible to form a well-mixed phase, which is favorable in ternary blend films[28,29].

## Device performance of binary and ternary OSCs

The impact of our molecular design on photovoltaic performance is investigated by devices with conventional structure of indium tin oxide (ITO)/poly (3,4-ethylenedioxythiophene): poly (styrene sulfonate) (PEDOT: PSS)/active layer/poly[[9,9-bis(3′-((N,N-dimethyl)-N-ethylammonium)-propyl)−2,7-fluorene)-alt-2,7-(9,9-dioctylfluorene)] dibromide (PFN-Br)/Ag. In this study, the classical polymer PM6 was selected as the donor material because of its complementary absorption (Fig. 2d) and suitable energy level (Fig. 2h)[5]. We employed additive 1,4-diodobenzene (DIB) to improve device performance, which we first demonstrated the eutectic phase behavior beneficial for NFA arrangement in 2021[30]. The $J$-$V$ curves of champion devices based on the two isomers are plotted in Fig. 3a, and the corresponding photovoltaic parameters are summarized in Table 2. o-BTP-eC9 based device shows a significantly higher $V_{OC}$ of 0.901 V than BTP-eC9 based device ($V_{OC}$ = 0.843 V) by 58 mV, which should be ascribed to the larger bandgap and less $V_{OC}$ loss of the PM6:o-BTP-eC9 device (will be discussed later). On the other hand, due to the narrower absorption range of o-BTP-eC9, the current density ($J_{SC}$) of o-BTP-eC9 based device ($J_{SC}$ = 26.33 mA/cm$^2$) is lower than that of the BTP-eC9 based device ($J_{SC}$ = 28.27 mA/cm$^2$). The external quantum efficiency (EQE) measurements (Fig. 3b) again verified the tendency from $J$-$V$ tests, with integrated $J_{SC}$ of 25.54 mA/cm$^2$ and 27.39 mA/cm$^2$ for the PM6:o-BTP-eC9 and PM6:BTP-eC9 based devices, respectively. Both devices show

excellent fill factor (FF) values of around 79%, and as a result, o-BTP-eC9 contributes a comparable champion PCE of 18.7% (Table 2), with obviously reduced energy loss (will be discussed later).

Considering the higher LUMO level and dielectric constant of o-BTP-eC9, as well as the good miscibility between o-BTP-eC9 and BTP-eC9, we introduced o-BTP-eC9 as the third component in the PM6:BTP-eC9 system to fine tune the micro morphology and energetics of the host blend. The photovoltaic parameters of ternary devices with different ratios of acceptors are summarized in Supplementary Table 4. As we can see, in the range of 10 wt% to 30 wt% for o-BTP-eC9, the $V_{OC}$ of device continuously increases, while $J_{SC}$ and FF undergo a process of first increasing and then decreasing. When 15 wt% BTP-eC9 was replaced by o-BTP-eC9, the ternary device showed a champion PCE of 19.88% (Table 2 and Fig. 3a), with a $V_{OC}$ of 0.860 V, a $J_{SC}$ of 28.75 mA/cm$^2$ (integrated $J_{SC}$ of 27.83 mA/cm$^2$ from EQE measurement) and an excellent FF of 80.41%. The device was further encapsulated and sent to an ISO/IEC 17025:2017 accredited calibration lab−Enli Tech Optoelectronic Calibration Lab. A certified efficiency of 19.48% (Fig. 3c and Supplementary Fig. 20) was rated by asymptotic maximum power ($P_{max}$) scan recommended by the National Renewable Energy Laboratory[31,32], which is more reliable than the typical fast current-voltage (I-V) scan. To the best of our knowledge, 19.9% efficiency (19.5% certified by Asymptotic $P_{max}$ Scan) is the record value for single-junction OSCs (Fig. 3d). In addition, the ternary OSC shows excellent operational stability under 1-sun illumination stress test at MPP (maximum power point). As presented in Supplementary Fig. 21, the two binary devices suffer from more serious burn-in loss, so the ternary OSC exhibits the higher $T_{80}$ (the time in which device efficiency drop to 80% of initial value) of 724 h than the PM6:BTP-eC9 ($T_{80}$ = 428 h) and PM6:o-BTP-eC9 ($T_{80}$ = 578 h) OSCs.

## Device physics

Compared to the host PM6:BTP-eC9 OSC, the ternary device displays improvements in all the three photovoltaic parameters ($V_{OC}$, $J_{SC}$ and FF). To get more insights into the role of o-BTP-eC9, we first analyzed

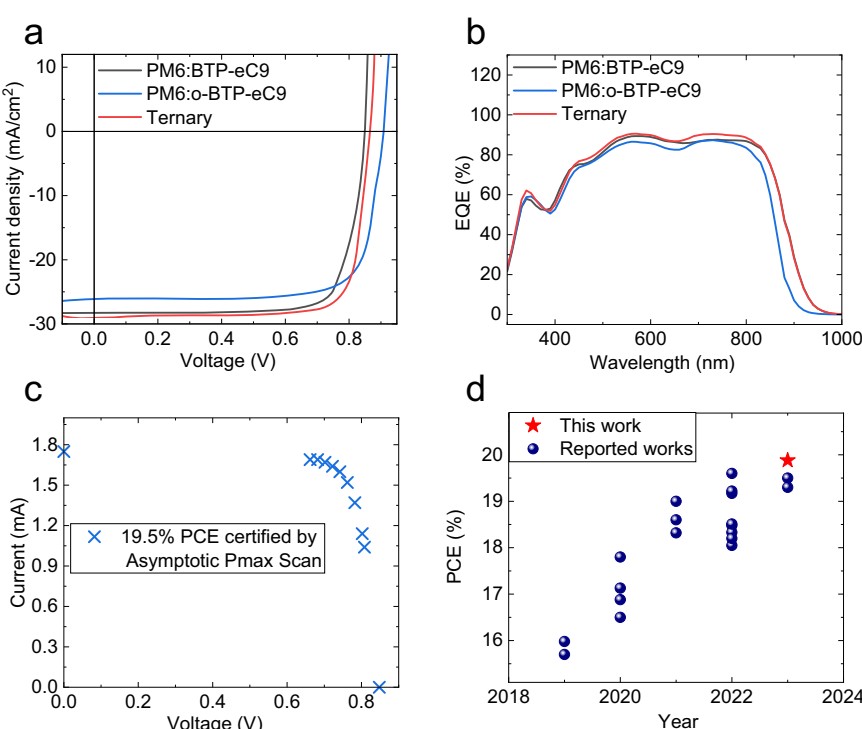

**Fig. 3 | Device performance.** $J$−$V$ curves (**a**) and EQE spectra (**b**) of the binary and ternary OSCs. **c** Certified results from Enli Tech. Optoelectronic Calibration Lab. **d** Comparison of PCEs versus year in efficient OSCs. Source data are provided as a Source Data file.

**Table 2 | Detailed photovoltaic parameters of binary and ternary devices**

| Active layer | $V_{OC}$ (V) | $J_{SC}$ (mA cm$^{-2}$) | $J_{SC}^{cal}$ (mA cm$^{-2}$) | FF (%) | PCE$^a$ (%) |
|---|---|---|---|---|---|
| PM6:BTP-eC9 | 0.843 | 28.27 | 27.39 | 79.17 | 18.87 (18.63 ± 0.11) |
| PM6:o-BTP-eC9 | 0.901 | 26.33 | 25.54 | 78.83 | 18.71 (18.50 ± 0.11) |
| Ternary | 0.860 | 28.75 | 27.83 | 80.41 | 19.88 (19.55 ± 0.15) |
| Ternary$^b$ | 0.848 | 28.78 | / | 79.86 | 19.48 |

$^{cal}$Integrated $J_{SC}$ values from EQE measurements.
$^a$The average PCEs with standard deviation calculated from 30 devices. All devices were tested with a metal mask applied.
$^b$The certified photovoltaic parameters from an accredited calibration lab, Enli Tech. Optoelectronic Calibration Lab. Accreditation Criteria: ISO/IEC 17025:2017.

**Table 3 | Detailed $E_{loss}$ parameters of binary and ternary devices**

| Active layer | $V_{OC}^a$ (V) | $E_{loss}$ (eV) | $E_g$ (eV) | $\Delta E_1$ (eV) | $\Delta E_2$ (eV) | EQE$_{EL}$ (%) | $\Delta E_3$ (eV) | $\Delta E_3^{cal}$ (eV) |
|---|---|---|---|---|---|---|---|---|
| PM6:BTP-eC9 | 0.855 | 0.541 | 1.396 | 0.261 | 0.069 | $4.0 \times 10^{-2}$ | 0.202 | 0.211 |
| PM6:o-BTP-eC9 | 0.913 | 0.500 | 1.413 | 0.263 | 0.048 | $7.0 \times 10^{-2}$ | 0.188 | 0.189 |
| Ternary | 0.872 | 0.525 | 1.397 | 0.262 | 0.064 | $6.6 \times 10^{-2}$ | 0.190 | 0.199 |

$^a$Devices were tested without metal mask applied.

the $V_{OC}$ loss of these binary and ternary devices. The $V_{OC}$ loss in OSCs consists of three components, radiative recombination above the bandgap ($\Delta E_1$), radiative recombination below the bandgap ($\Delta E_2$) and non-radiative recombination loss ($\Delta E_3$ or $E_{loss, nr}$)[33]. As $\Delta E_1$ is unavoidable and only determined by temperature and the bandgap of light-absorbing material[34], the key to $V_{OC}$ improvement is to strive for a decrease in $\Delta E_2$ and $\Delta E_3$. The origin of $\Delta E_2$ is the redshifted absorption caused by the existence of charge transfer (CT) state[35,36]. As for $\Delta E_3$, according to reciprocity theory, it can be calculated from the electroluminescence quantum efficiency (EQE$_{EL}$) of the device, the equation is expressed as $\Delta E_3 = -\frac{kT}{q} \ln EQE_{EL}$, where $k$ is the Boltzmann constant, $T$ is the Kelvin temperature and $q$ is the elementary charge[37]. To be specific, higher EQE$_{EL}$ value corresponds to lower $\Delta E_3$ and an ideal photovoltaic device without non-radiative recombination loss should also be a perfect light emitting diode (LED) with 100% EQE$_{EL}$.

The calculation process is presented in Supplementary Information, and the detailed $V_{OC}$ loss parameters are summarized in Table 3 and Fig. 4a. The bandgap ($E_g$) of the device was determined with a probability distribution method[38,39], as shown in Supplementary Fig. 22, the determined $E_g$s of PM6:BTP-eC9 device, PM6:o-BTP-eC9 device, and the ternary device are 1.396, 1.413, and 1.397 eV, respectively. By subtracting the $V_{OC}$, the corresponding $E_{loss}$ values are 0.541, 0.500, and 0.524 eV, respectively. Specifically, these three types of devices showed similar $\Delta E_1$ of 0.26 eV. But the $\Delta E_2$ of PM6:o-BTP-eC9 device ($\Delta E_2 = 0.048$ eV) is obviously smaller than that of PM6:BTP-eC9 device ($\Delta E_2 = 0.069$ eV). We believe the reasons for the reduced $\Delta E_2$ in PM6:o-BTP-eC9 device are two-fold. One is the higher $E_{CT}$ (the energy of CT state) caused by the changes in chlorination position, as demonstrated in Fig. 4b, c. Another one is the higher dielectric constant of o-BTP-eC9, which facilitates exciton dissociation and reduce recombination loss in OSC[23]. The $\Delta E_2$ of the ternary device lies between that of two binary devices, implying the incorporation of the guest acceptor o-BTP-eC9 has a positive impact on improving the $E_{CT}$ of the ternary device (Fig. 4d). As for the calculation of $\Delta E_3$, there are two widely used methods. One is calculated from the EQE$_{EL}$, as mentioned in the previous paragraph:$\Delta E_3 = -\frac{kT}{q} \ln EQE_{EL}$. Here, the benchmark PM6:BTP-eC9 device showed the weakest EQE$_{EL}$ of $4.0 \times 10^{-4}$ (Fig. 4e), corresponding to the highest $\Delta E_3$ of 0.202 eV. In comparison, the EQE$_{EL}$ of PM6:o-BTP-eC9 device is 75% higher, with a value of $7.0 \times 10^{-4}$ ($\Delta E_3 = 0.188$ eV). The optimized ternary device shows EQE$_{EL}$ of $6.6 \times 10^{-4}$ ($\Delta E_3 = 0.190$ eV), 65% higher than that of benchmark PM6:BTP-eC9 device. It is worth noting that the $\Delta E_3$ values of 0.188 and 0.190 eV are among the lowest non-radiative recombination loss in

OSCs with over 18% PCE (Fig. 4f). Another method used to calculate $\Delta E_3$ is from $J$-$V$ characteristic, following the equation of $\Delta E_3^{cal} = E_g - qV_{OC} - \Delta E_1 - \Delta E_2$. Like the tendency observed in the first method, PM6:BTP-eC9 device suffers from more serious $\Delta E_3$ (0.211 eV) than that of PM6:o-BTP-eC9 device ($\Delta E_3 = 0.189$ eV) and ternary device ($\Delta E_3 = 0.199$ eV). The lowest $\Delta E_3$ in the PM6:o-BTP-eC9 system should be ascribed to the lower LUMO energy level offset, which contributes to the hybrid of singlet excited state (S1) and CT state, and thus improves device EQE$_{EL}$[20,21]. Overall, the molecular design can obviously reduce the radiative and non-radiative recombination in OSC, therefore the PM6:o-BTP-eC9 device exhibited a high efficiency of 18.7% although with a lower $J_{SC}$ than the benchmark PM6:BTP-eC9 device. Due to the incorporation of o-BTP-eC9, the ternary device also showed lower $\Delta E_2$ and $\Delta E_3$ than the host PM6:BTP-eC9 binary device.

Except for $V_{OC}$ loss, we also systematically investigated the charge transport, recombination, and collection processes of these three OSC systems. As shown in Supplementary Fig. 23 and Supplementary Table 5, we used space charge limited current (SCLC) method to estimate the charge mobilities of binary and ternary OSCs. The hole mobilities ($\mu_h$) of all these three devices are similar, around $4 \times 10^{-4}$ cm$^2$/Vs. While the electron mobility ($\mu_e$) of PM6:BTP-eC9 device ($\mu_e = 5.1 \times 10^{-4}$ cm$^2$/Vs) is faster than that of PM6:o-BTP-eC9 device ($\mu_e = 3.5 \times 10^{-4}$ cm$^2$/Vs), which should be ascribed to the higher crystallinity of BTP-eC9. Due to the introduction of o-BTP-eC9, the ternary device shows slower electron mobility than the host PM6:BTP-eC9 device, but the ratio of $\mu_h$ and $\mu_e$ of ternary device is closer to 1 ($\mu_h = 3.8 \times 10^{-4}$ cm$^2$/Vs, $\mu_e = 4.0 \times 10^{-4}$ cm$^2$/Vs). It is well accepted that the balanced charge transport is beneficial to reducing space current and thereby improving photocurrent and FF.

We then performed transient photocurrent (TPC) and transient photovoltage (TPV) measurements to study the charge dynamics of binary and ternary OSCs. By fitting the current and voltage curves with mono-exponential decay model, we can get the corresponding decay time $\tau$[40]. TPC testes were operated under short-circuit condition, the larger $\tau$ value means the less efficient charge extraction in device. As presented in Supplementary Fig. 24a, the ternary OSC shows the shortest $\tau$ of 0.66 μs, illustrating the guest o-BTP-eC9 effectively accelerates charge extraction process. TPV tests were conducted under open-circuit condition, the smaller $\tau$ value means the more serious recombination process in device. Supplementary Fig. 24b shows the TPV results, and as we can see, the ternary device shows the longest $\tau$ of 1.97 μs, indicating the introduction of o-BTP-eC9 is an effective method to restrict charge recombination. As shown in

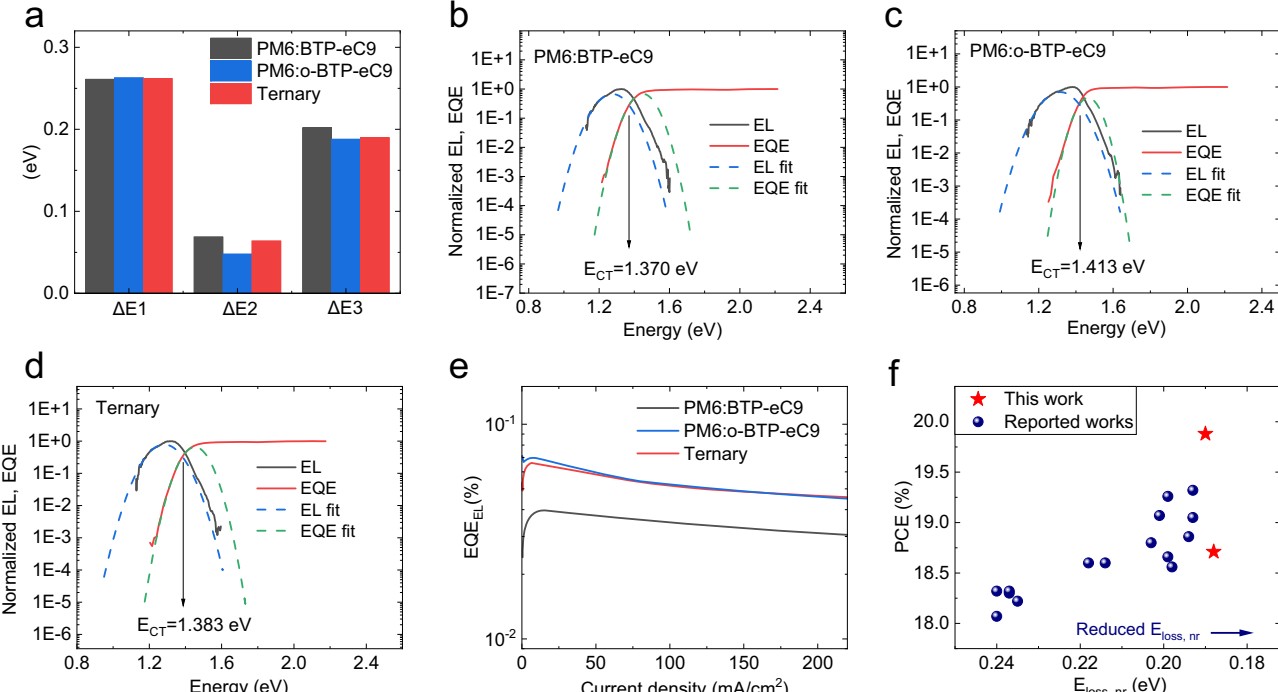

**Fig. 4 | Device physics. a** The comparison of detailed energy loss for binary and ternary OSCs. $E_{CT}$ determination for the PM6:BTP-eC9 based (**b**), PM6:o-BTP-eC9 based (**c**) and ternary (**d**) OSCs, here EL represents the electroluminescence. **e** EQE$_{EL}$ spectra for binary and ternary OSCs. **f** Comparison of PCEs versus non-radiative recombination values in reported OSCs with over 18% efficiency. Source data are provided as a Source Data file.

Supplementary Fig. 25, we further studied the trap-assisted recombination, also known as Shockley-Read-Hall recombination, by measuring the dependence of $V_{OC}$ on light intensity ($P_{light}$). The relation of $V_{OC} \propto nkT/q \ln(P_{light})$ can be used to evaluate the situation of trap-assisted recombination, where $n$, $k$, $T$, and $q$ are the fitted slope of $V_{OC}$ versus $P_{light}$ on a logarithmic scale, the Boltzmann constant, the temperature of Kelvin, and the elementary charge, respectively[41,42]. The fitted slope of ternary device is 1.02, lower than the values of two binary devices, which indicates that the trap-assisted recombination in the ternary device is suppressed by o-BTP-eC9.

To estimate the charge collection process, we measured the photocurrent density ($J_{ph}$) versus the effective voltage ($V_{eff}$) of binary and ternary devices. $J_{ph}$ can be defined as $J_L - J_D$, where $J_L$ and $J_D$ are the current densities under illumination and in the dark, respectively[43,44]. $V_{eff}$ is equal to $V_0 - V_{bias}$, where $V_0$ is the voltage when $J_{ph} = 0$, and $V_{bias}$ is the applied voltage bias. As shown in Supplementary Fig. 26, $J_{ph}$ becomes saturated when $V_{eff}$ increases to 2 V. We then calculated the charge collection probability ($P_{coll}$) according to the equation $P_{coll} = J_{ph}/J_{sat}$. At short-circuit condition, the ternary OSC showed the highest $P_{coll}$ value of 99.2%, agreeing well with the TPC results. Combination of the more balanced charge transport, suppressed recombination and more efficient charge collection contributes to the enhanced $J_{SC}$ and excellent FF in ternary devices.

**Nano-morphology and molecular packing in binary and ternary blend films**

To understand the relationship between device performance and phase-separated morphology, tapping-mode atomic force microscopy (AFM) measurements were conducted. Figure 5a–c are the height images of the PM6:BTP-eC9, PM6:o-BTP-eC9, and the ternary blend films, respectively. All three blend films show delicate nanomorphology, but due to the high crystallinity of BTP-eC9, the PM6:BTP-eC9 film shows the most obvious molecular aggregation, with the highest root-mean-square roughness (Rq) of 0.99 nm. After introducing the less crystalline o-BTP-eC9, the excessive molecular aggregation in the host

PM6:BTP-eC9 blend is suppressed, with the smaller Rq of 0.70 nm in the ternary film. The phase-separated nanostructures are more obvious in Fig. 5d–f, all films exhibit visible fibrillar networks, which is beneficial to exciton dissociation and charge transport. The favorable phase-separated morphology should result from the suitable miscibility between the donor and acceptor materials and should be a reason for the excellent FF values in all three devices. But like the phenomenon observed in height images, the PM6:BTP-eC9 film shows obviously larger phase clusters than other two films due to the higher molecular aggregation. To get a quantitative analysis, we plot line profiles of phase images in Supplementary Fig. 27, the PM6:BTP-eC9 film shows the largest average phase cluster size of 18.1 nm. After the incorporation of o-BTP-eC9, the phase clusters of the ternary film become more uniform and the average size (15.7 nm) become smaller, which is more favorable for exciton diffusion, exciton dissociation and charge carrier transport. The AFM images imply that the crystallinity difference and good miscibility between BTP-eC9 and o-BTP-eC9 fine-tuned the aggregation of NFA, thereby improving the morphology in the ternary film.

We further investigated the influence of o-BTP-eC9 on molecular stacking in BHJ by GIWAXS measurements, and the detailed information of lamellar diffraction peaks and π-π stacking (010) diffractions is summarized in Supplementary Table 6. Figure 6a–c presents the two-dimensional (2D) GIWAXS patterns of the PM6:BTP-eC9, PM6:o-BTP-eC9 and ternary blend films, respectively. Along the in-plane direction (Fig. 6d), all blend films show two lamellar diffraction peaks, ascribing to the lamellar stacking of PM6 ($q$ at around 0.3 Å$^{-1}$) and NFAs ($q$ at around 0.4 Å$^{-1}$), respectively. Along the out-of-plane (OOP) direction (Fig. 6e), there are distinct π-π diffractions ($q$ at around 1.7 Å$^{-1}$) for all films, implying these three blends all show preferred π-π stacking. But compared to the PM6:BTP-eC9 blend, the PM6:o-BTP-eC9 blend present obviously lower diffraction intensity due to the crystallinity difference of o-BTP-eC9. Like the tendency observed in the GIWAXS patterns of neat films, the two binary blend films show similar π-π stacking distance of 3.7 Å, but the CCL$_{010}$ (22.5 Å) of PM6:o-BTP-eC9

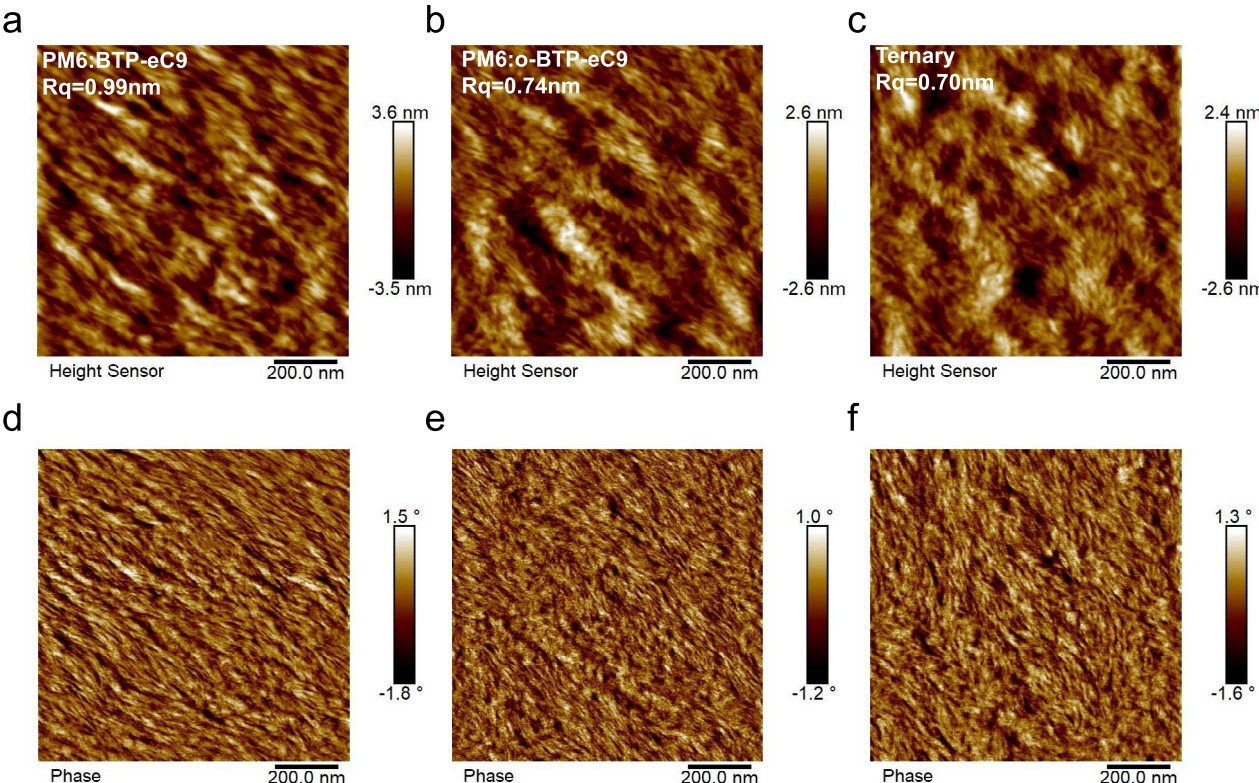

**Fig. 5 | Surface topography of binary and ternary blend films.** AFM height images of the PM6:BTP-eC9 (**a**), PM6:o-BTP-eC9 (**b**) and the ternary (**c**) blend films. AFM phase images of the PM6:BTP-eC9 (**d**), PM6:o-BTP-eC9 (**e**) and ternary (**f**) blend films.

blend is smaller than that (26.4 Å) of PM6:BTP-eC9 blend. Because of the crystallinity difference between these two isomers, the $CCL_{010}$ value of the ternary film lies between that of two binary films. The role of o-BTP-eC9 in moderating the crystallization of BTP-eC9 was reconfirmed by grazing incidence small-angle X-ray scattering (GISAXS) measurements. Figure 6g–i are the 2D GISAXS patterns of binary and ternary films, and Fig. 6h is the corresponding 1D profiles along the IP direction. An 1D GISAXS profile consists of a shoulder in the middle-q region, an upturn in the small-q region and an almost straight line in the high-q region. The shoulder originates from the crystalline regions of the blend, and the blunter shoulder corresponds to the larger size of crystalline region[45,46]. To get a quantitative comparison, we fitted these profiles by a fractal network model. The fitted average sizes of crystalline region in the PM6:BTP-eC9, PM6:o-BTP-eC9 and ternary blend films are 21, 12, and 18 nm, respectively, again verifying the molecular crystallization of the host blend is effectively optimized by incorporating the guest component. The AFM, GIWAXS and GISAXS measurements jointly demonstrate that the complementary crystallinity and good miscibility between BTP-eC9 and o-BTP-eC9 contributes to a more ideal phase separation morphology.

## Discussion

In conclusion, theoretical DFT calculation guided the design and synthesis of a promising NFA by changing the chlorination position in IC-2Cl groups. The o-BTP-eC9 molecule shows shallower LUMO level, higher dielectric constant, and weaker crystallinity than BTP-eC9. The upshifted CT state in the PM6:o-BTP-eC9 OSC leads to suppressed radiative and non-radiative recombination. More excitingly, benefiting from the compatibility of the two NFAs in crystallinity and energy level, o-BTP-eC9 not only optimizes the energetics of ternary OSC but also fine tunes the phase separation morphology of the host PM6:BTP-eC9 blend. As a result, in the PM6:BTP-eC9:o-BTP-eC9 based ternary OSC, a record 19.9% PCE of single junction OSC with reduced energy loss was

achieved. This work paves an avenue for designing new NFA guest molecules toward high performance OSCs.

## Methods

### Materials

All materials are provided by commercial suppliers: PEDOT:PSS (Clevios P VP AI. 4083 (Heraeus)), PM6 (Solarmer Energy Inc.), BTP-eC9 (Solarmer Energy Inc.), BTP-eC9-CHO (Jiangsu ji'a Biotechnology Co., Ltd.), 2,3-Dichlorobenzoic Acid (Shanghai Titan Scientific Co., Ltd.), DIB (Tokyo Chemical Industry Co., Ltd.), PFN-Br (Solarmer Energy Inc.), Chloroform (Sigma-Aldrich, Ltd.), and isopropyl alcohol (Sigma-Aldrich, Ltd.), Ag pellet (ZhongNuo Advanced Material (Beijing) Technology Co., Ltd.). And all reagents and solvents are used directly without further purification.

### Device fabrication and testing

The OSCs were fabricated with a conventional structure of ITO/PEDOT:PSS/active layer/PFN-Br/Ag. At first, the ITO-coated glass substrates were cleaned sequentially with detergent, de-ionized water, acetone, and isopropyl alcohol (IPA) for 15 min under sonication. Then the substrates were dried in nitrogen flow and treated with UV ozone for 30 min. After that, about 50 ul PEDOT:PSS was dripped on ITO substrates and spin-coated at 6000 rpm for 20 s, followed by thermal annealing on a hot plate at 120 °C for 10 min to remove the water in PEDOT:PSS film. Then, the substrates were transferred into a glovebox filled with nitrogen ($O_2 < 10$ ppm; $H_2O < 10$ ppm). The precursor solutions were prepared by mixing 7.3 mg donor, 8.7 mg acceptor(s), and 12.5 mg DIB in 1 ml chloroform. For the optimized ternary device, the weight ratio of BTP-eC9: o-BTP-eC9 is 1.05:0.15. And the precursor solution would be heated to 60 °C for an hour to fully dissolve the solutes then cooled to room temperature before use. The thickness of the active layer was controlled at around 120 nm, then the active layer experienced a process of thermal annealing at 100 °C for 5 min. The

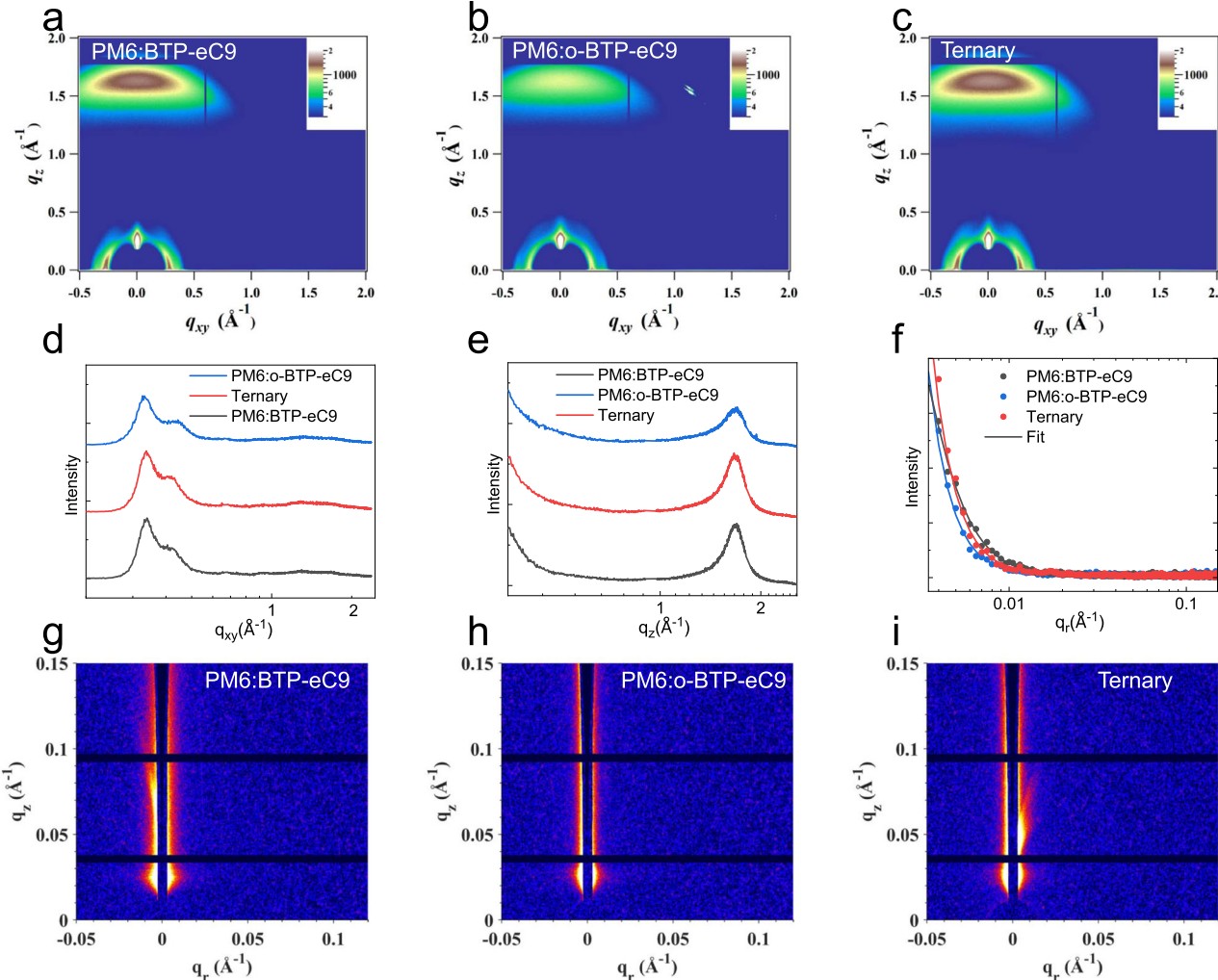

**Fig. 6 | GIWAXS and GISAXS measurements.** 2D GIWAXS diffraction patterns of the PM6:BTP-eC9 (**a**), PM6:o-BTP-eC9 (**b**) and ternary (**c**) blend films. The related 1D GIWAXS line cut profiles along IP (**d**) and OOP (**e**) directions. **f** 1D GISAXS profiles along $q_r$ direction. 2D GISAXS diffraction patterns of the PM6:BTP-eC9 (**g**), PM6:o-BTP-eC9 (**h**) and ternary (**i**) blend films. Source data are provided as a Source Data file.

next stage is to coat electron transport material, about 5 nm PFN-Br (0.5 mg/ml in methanol) was coated on the top of the active layer. Finally, these semi-finished cells were transferred into a thermal evaporation chamber with a base pressure of ~2 × 10⁻⁴ Pa, where 100 nm Ag was deposited through a shadow mask with the active area of 11 mm². The *J–V* curves of OSCs were tested by a Keithley 2400 source meter and an AAA grade solar simulator (SS-F7-3A, Enli Tech. Co., Ltd., Taiwan) along with AM 1.5 G spectra whose intensity was corrected by a standard silicon solar cell at 1000 W/m². The *J–V* curves are measured in the forward direction from −0.2 to 1.2 V, and all devices were covered with anti-reflection films (Mitsubishi Chemical Corporation) when conducted *J–V* test. The external quantum efficiency (EQE) was measured by a certified incident photon to electron conversion (IPCE) equipment (QE-R) from Enli Technology Co., Ltd.

**AFM, GIWAXS and GISAXS**
AFM images were acquired using a Bruker Dimension EDGE in tapping mode. The GIWAXS data was obtained from the PLS II 3C SAXS-I and 9A U-SAXS beamline of the Pohang Accelerator Laboratory in Korea. The active layer samples were deposited on the Si/PEDOT:PSS substrates following device conditions. The X-rays coming from the in-vacuum undulator (IVU) were monochromated (wavelength $\lambda = 1.10994$ Å) using a double crystal monochromated and focused both horizontally

and vertically (100 (H) × 20 (V) μm² in FWHM @ the sample position) using K-B type mirrors. The grazing incidence wide-angle X-ray scattering sample stage was equipped with a 7-axis motorized stage for the fine alignment of the sample, and the incidence angles of the X-beam was set to be 0.11°–0.13°. The GIWAXS patterns were recorded with a 2D CCD detector (Rayonix SX165) and an X-ray irradiation time within 100 s, dependent on the saturation level of the detector. Diffraction angles were calibrated using a sucrose standard (monoclinic, P21, $a = 10.8631$ Å, $b = 8.7044$ Å, $c = 7.7624$ Å, and $\beta = 102.938°$) and sample-to-detector distance was ~231 mm. The GISAXS measurements were performed with a Xeuss 2.0 SAXS/WAXS laboratory beamline using a Cu X-ray source (8.05 keV, 1.54 Å) and a Pilatus3R 300 K detector.

**Highly sensitive EQE and EQE_EL measurements**
Highly sensitive EQE was measured using an integrated system (PECT-600, Enlitech), where the photocurrent was amplified and modulated by a lock-in instrument. EQE_EL measurements were performed by applying external voltage/current sources through the devices (ELCT-3010, Enlitech).

**Reporting summary**
Further information on research design is available in the Nature Portfolio Reporting Summary linked to this article.

## Data availability
The data that support the findings of this study are presented in the main text and Supplementary Information file. All the data are available from the corresponding authors on request. The X-ray crystallographic coordinates for structures reported in this study have been deposited at the Cambridge Crystallographic Data Centre (CCDC), under deposition numbers 2312645. These data can be obtained free of charge from The Cambridge Crystallographic Data Centre via www.ccdc.cam.ac.uk/data_request/cif. Source data are provided with this paper.

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

## Acknowledgements

G.L. acknowledges the support from Research Grants Council of Hong Kong (Project Nos. 15221320, C5037-18G), RGC Senior Research Fellowship Scheme (SRFS2223-5S01), Hong Kong Innovation and Technology Commission (GHP/205/20SZ), Shenzhen Science and Technology Innovation Commission (JCYJ20200109105003940), the Hong Kong Polytechnic University Internal Research Funds: (Sir Sze-yuen Chung Endowed Professorship Fund (8-8480), G-SAC5), and Guangdong-Hong Kong-Macao Joint Laboratory for Photonic-Thermal-Electrical Energy Materials and Devices (GDSTC No. 2019B121205001). This work was financially supported by research grants from Natural Science Foundation of China (22071238), Special Research Assistant Program of Chinese Academy of Sciences (E2296108), and Chongqing Postdoctoral Science Foundation (cstc2021jcyj-bshX0062). Portions of this research were carried out at the 3C and 9A beam lines of the Pohang Accelerator Laboratory, Republic of Korea.

## Author contributions

J.F., Q.Y., Z.X., and G.L. conceived this work. J.F. fabricated the devices and performed data analysis. Q.Y. synthesized the small molecular acceptor o-BTP-eC9 and performed the UV-vis, CV, and AFM measurements. P.H. conducted quantum chemistry calculation. S.C., K.C. and Z.K. performed GIWAXS measurements. He.L. and X.L. performed GISAXS measurements. Y.L., Ha.L., and F.H. conducted the single crystal analysis. P.F. maintained the equipment and gave many useful suggestions on device fabrication. J.F. and Q.Y. wrote the manuscript. Z.X., S.L., Y.Y., and G.L. directed the subject and revised the manuscript. All authors discussed the results and commented on the manuscript.

## Competing interests

The authors declare no competing interests.
