## [Peer Review File · Nature Communications]

Rational molecular and device design enables organic solar cells approaching 20% efficiencyREVIEWER COMMENTS

Reviewer #1 (Remarks to the Author):

In this manuscript, Li et al developed a new NFA naming o-BTP-eC9 by changing the chlorination position in IC-2Cl groups. This molecule shows lower synthetic complexity, shallower LUMO level, higher dielectric constant, and weaker crystallinity than the well-known BTP-eC9. Benefiting from the compatibility of the o-BTP-eC9 and BTP-eC9 in crystallinity and energy level, the PM6:BTP-eC9:o-BTP-eC9 based ternary OSC obtained a record 19.9% PCE of single junction OSC with reduced energy loss. This work provided new idea for designing new NFA molecules towards both low-cost and high performance OSCs simultaneously. Thus I recommend it for publication with following comments:

1. The procedure for the synthesis of o-BTP-eC9 in SI has a lot of mistakes. "Thionyl chloride (4.4 ml g)" should be changed. "ml" should be changed to "mL". "(5.7303 g, 30 mmol)" should be "(5.73 g, 30 mmol)". "(4.6717 g, yield72.42%)" should be "(4.67 g, yield: 72.4%)".
2. For NMR format, it also has a lot of mistakes. Please correct it. For example, "J" should be italic.
3. Coupling constant (J) should be given for the ¹H NMR of o-BTP-eC9 in SI.
4. Please zoom the overlapping area of ¹³C NMR spectra in SI.
5. Figure S10, it seems that the absorption has been normalized then the Y axis should be "normalized absorption".
6. For PM6:o-BTP-eC9 device, the Voc is significantly improved but the Jsc decreased obviously. The possible reason should be given in the main text.
7. How about the performance for the D18: o-BTP-eC9 device?

Reviewer #2 (Remarks to the Author):

This manuscript describes the synthesis of a novel NFA (o-BTP-eC9) guided by theoretical considerations. Organic solar cells based on the well-known donor PM6 and the reference NFA BTP-eC9 including ternary cells using all three components were fabricated. The optimal ternary blend cell showed a record PCE for single junction OPV of 19.9% (19.5% certified). The authors have executed a very thorough study including detailed photophysical, morphology, and stability studies.

The main concern is regarding the claims about reduced synthetic complexity (SC). The authors have shown with an established equation that the SC is reduced and is low compared to a variety of materials. However, if comparing o-BTP-eC9 directly to BTP-eC9, the number of synthetic steps is only reduced by one from 14 to 13. The overall yield is higher at 5.1% vs. 1.8%. However, this is really splitting hairs. The

authors have not demonstrated a novel approach to generating simplified NFAs with reduced synthetic complexity. This is just one more Y-series NFA that takes more than 10 steps to make and has a very low overall yield. In the opinion of this reviewer, there is no real advance on this point.

The real advance is in the rational design and the execution in devices that leads to higher efficiency. Focusing on lower synthetic complexity and claiming an advance on this aspect is disingenuous and misleading and shows a real loop-hole in the equations used to calculate and represent SC. This paper is very strong without this line of discussion or these SC claims, they are not needed and actually detract from the quality of the paper. This reviewer is of the opinion that these claims and this line of discussion should be removed. The paper is otherwise very strong and can stand on its own.

Reviewer #3 (Remarks to the Author):

The paper reports the synthesis of a new non-fullerene acceptor, for use in organic photovoltaic devices. The design is a very small tweak on the existing NFA materials, in that the terminal end-group has a different isomer pattern of its two chlorine groups. There has been significant work on the nature of end-group isomers, and indeed this exact same molecule has already been published <https://onlinelibrary.wiley.com/doi/full/10.1002/adv.202207678> (May 12th) by one of the authors of this study (Shirong Lu). The only difference is the recent paper used a C11 sidechain on the central core, whereas this paper uses a C9.

The synthesis of the endgroup and its simplification compared to other dechlorinated end-groups are discussed in this paper (in a very similar manner to the current manuscript), and the dipole moment is also calculated and presented.

A synthetic complexity analysis is also presented in the published paper, as in the current manuscript. Interestingly, the same materials get a different SC score in each paper. For example, BTP-eC9 is 92.6 in the Wiley paper, but (table SS1) 79 here. Similar differences occur for other materials in each manuscript. Given that these are based on published analysis, the reason is not clear.

Furthermore, for the synthetic complexity comparison of BTP-eC9, versus the current NFA; I find the different values surprising given the only change is the end group. Table SS1 reports ref 11 for the synthesis of BTP-eC9, although no details are given so one must go back to https://onlinelibrary.wiley.com/doi/epdf/10.1002/adma.201703080?saml_referrer for the synthesis of the end group. This only involves one column chromatography step as far as I can see. The reported synthesis of the new isomer involves 2 CC steps (as reported by the authors). However, table SS1, gives both materials as only 6 CC. Since the core is the same, there must be an error somewhere. Please check.

More fundamentally, the number from this analysis must only be used very broadly, and quoting to two or even decimal place seems meaningless to me. The analysis is only based on the small-scale research synthesis, which is unlikely to be used for scale-up.

For the synthesis of the new endgroup, only ¹H and ¹³C is provided to prove structure. Please also include other relevant characterization to prove structure, such as mass spectra, melting point.

The authors comment in comparing to BTP-eC9, that 'due to the orientation effect of 2,3-dichlorobenzoyl chloride, o-BTP-eC9 136 needs fewer synthetic steps, and has much higher reciprocity yield (5.1% versus 1.8%) 137 than BTP-eC9 (Table SS1 in Supplementary Information 2).] This makes little sense to me – the only difference is the end-group. The authors prepare there end group in two steps, with a overall yield of 54%. The alternative diCl end group is prepared in 3 three steps in 36% yield (with only one chromatography step). This is a better comparison. The total synthesis of the NFA is the same from that point one, and any differences in yield likely relate to user experience.

Please report the optical absorption coefficient, both in solution and in thin-film for the new acceptor. This will help to further prove the increase in dielectric constant reported for the new acceptor.

Fig 3 – specify what FOM is being used.

Overall, clearly the selling point of the paper is the high efficiency of the blends, approaching record values, but the novelty in terms of molecular design is low, not least in light of the Adv. Sci paper the authors have recently submitted, which has significant overlaps with the current manuscript (and some contradictions in terms of SC analysis). Therefore, I think the current manuscript needs to be updated to address and acknowledge the Adv. Sci paper, and the SC needs to be carefully checked. After these changes, the paper may be suitable for this journals.

(The changes are highlighted in red color in the revised manuscript. The comments by the editor and reviewers are in black color and our responses are in blue color.

Reviewer #1 (Remarks to the Author):

In this manuscript, Li et al developed a new NFA naming o-BTP-eC9 by changing the chlorination position in IC-2Cl groups. This molecule shows lower synthetic complexity, shallower LUMO level, higher dielectric constant, and weaker crystallinity than the well-known BTP-eC9. Benefiting from the compatibility of the o-BTP-eC9 and BTP-eC9 in crystallinity and energy level, the PM6:BTP-eC9:o-BTP-eC9 based ternary OSC obtained a record 19.9% PCE of single junction OSC with reduced energy loss. This work provided new idea for designing new NFA molecules towards both low-cost and high performance OSCs simultaneously. Thus I recommend it for publication with following comments:

Response:

We thank the reviewer for the positive comment on our work.

1. The procedure for the synthesis of o-BTP-eC9 in SI has a lot of mistakes. “Thionyl chloride (4.4 ml g” should be changed. “ml” should be changed to “mL”. “(5.7303 g, 30 mmol)” should be “(5.73 g, 30 mmol)”. “(4.6717 g, yield72.42%)” should be “(4.67 g, yield: 72.4%)”.

Response:

We thank the reviewer for pointing out this problem.

These typos have been corrected in the revised manuscript.

2. For NMR format, it also has a lot of mistakes. Please correct it. For example, “J” should be italic.

Response:

We thank the reviewer for pointing out these typos.

The related format problem has been addressed in the revised manuscript.

3. Coupling constant (J) should be given for the ¹H NMR of o-BTP-eC9 in SI.

Response:

We thank the reviewer for the comment.

The constant has been added accordingly.

4. Please zoom the overlapping area of ¹³C NMR spectra in SI.

Response:

We thank the reviewer for the comment.

The overlapping area of ^{13}C NMR spectra has been zoomed.

5. Figure S10, it seems that the absorption has been normalized then the Y axis should be “normalized absorption”.

Response:

We thank the reviewer for pointing out this problem.

Y-axis titles in Figure S10 have been changed to “Normalized absorption (a.u.)”.

6. For PM6:o-BTP-eC9 device, the V_{oc} is significantly improved but the J_{sc} decreased obviously. The possible reason should be given in the main text.

Response:

We thank the reviewer for the comment.

There are two reasons for the improved V_{oc} and decreased J_{sc} of the PM6:o-BTP-eC9 device. The first one is that the PM6:o-BTP-eC9 device has a larger bandgap than its PM6:BTP-eC9 counterpart (1.413 eV versus 1.397 eV, as shown in **Supplementary Fig. 21** and **Table 3**), meaning the light absorption range of the PM6:o-BTP-eC9 device is narrower than that of the PM6:BTP-eC9 device. The second reason is that the PM6:o-BTP-eC9 device suffers from less V_{oc} loss than the PM6:o-BTP-eC9 counterpart (0.500 eV versus 0.541 eV, as shown in **Fig. 4** and **Table 3**), due to the reduced radiative and non-radiative recombination loss.

Correspondingly, the revised manuscript has added a paragraph explaining the improved V_{oc} and decreased J_{sc} of the PM6:o-BTP-eC9 device.

“o-BTP-eC9 based device shows a significantly higher V_{oc} of 0.901 V than BTP-eC9 based device (V_{oc} =0.843 V) by 58 mV, which should be ascribed to the larger bandgap and less V_{oc} loss of the PM6:o-BTP-eC9 device (will be discussed later). On the other hand, due to the narrower absorption range of o-BTP-eC9, the current density (J_{sc}) of o-BTP-eC9 based device (J_{sc} =26.33 mA/cm²) is lower than that of the BTP-eC9 based device (J_{sc} =28.27 mA/cm²).”

7. How about the performance for the D18: o-BTP-eC9 device?

Response:

We thank the reviewer for the comment.

We have fabricated the D18: o-BTP-eC9 device, as shown in **Fig. R1**, it shows a champion PCE of 17.51%, with a higher V_{oc} of 0.911 V than the PM6: o-BTP-eC9 device, but lower J_{sc} (25.60 mA cm⁻² from J-V test and 24.91 mA cm⁻² from EQE test) and FF (75.12%) values.

Fig. R1 Device performance of D18: o-BTP-eC9 OSC. $J-V$ (a) and EQE (b) curves of the D18: o-BTP-eC9 device.

Reviewer #2 (Remarks to the Author):

This manuscript describes the synthesis of a novel NFA (o-BTP-eC9) guided by theoretical considerations. Organic solar cells based on the well-known donor PM6 and the reference NFA BTP-eC9 including ternary cells using all three components were fabricated. The optimal ternary blend cell showed a record PCE for single junction OPV of 19.9% (19.5% certified). The authors have executed a very thorough study including detailed photophysical, morphology, and stability studies.

Response:

We thank the reviewer for the positive comment on our work.

The main concern is regarding the claims about reduced synthetic complexity (SC). The authors have shown with an established equation that the SC is reduced and is low compared to a variety of materials. However, if comparing o-BTP-eC9 directly to BTP-eC9, the number of synthetic steps is only reduced by one from 14 to 13. The overall yield is higher at 5.1% vs. 1.8%. However, this is really splitting hairs. The authors have not demonstrated a novel approach to generating simplified NFAs with reduced synthetic complexity. This is just one more Y-series NFA that takes more than 10 steps to make and has a very low overall yield. In the opinion of this reviewer, there is no real advance on this point.

The real advance is in the rational design and the execution in devices that leads to higher efficiency. Focusing on lower synthetic complexity and claiming an advance on this aspect is disingenuous and misleading and shows a real loop-hole in the equations used to calculate and represent SC. This paper is very strong without this line of discussion or these SC claims, they are not needed and actually detract from the quality of the paper. This reviewer is of the opinion that these claims and this line of discussion should be removed. The paper is otherwise very strong and can stand on its own.

Response:

We sincerely appreciate the reviewer's perspective and understand the concerns

regarding the reduced synthetic complexity (SC).

We agree that the main progress of this work lies in the rational design and execution of devices, which result in higher efficiency. Following the reviewer's advice, to avoid misleading and detracting from the main advancement, we have removed the discussions and claims on synthetic complexity.

Correspondingly, the related sections have been changed as below:
Supplementary Information 2 has been removed, and the relevant synthesis information (steps and yields) has been referred to reference 19 (Adv. Sci. 2023, 10, 2207678).

“high figure-of-merit” has been removed from the title of this manuscript

In the Abstract section, " we present a rational design of non-fullerene acceptor (NFA) o-BTP-eC9, with much lower synthetic complexity and distinct photoelectric properties ...", "much lower synthetic complexity and" has been deleted.

In the Introduction section, " Reducing the synthetic complexity of NFAs requires fewer synthetic steps, cheaper starting materials, and higher yields.", " Furthermore, our device exhibits the highest figure-of-merit ($FOM = \frac{PCE}{SC_{D:A}}$, $SC_{D:A}$ is the synthetic complexity of donor and acceptor materials^{20,21}) among the state-of-the-art OSCs..", and " lower synthetic complexity" have been deleted.

In the Results and Discussion sections, " We calculated the synthetic complexity (the equation is presented in Supplementary Information 2), a parameter used to evaluate the cost of an organic material, the lower synthetic complexity indicates the lower cost^{20,21,24}. As shown in Table S1 and Fig.2d, o-BTP-eC9 exhibits lower synthetic complexity than other efficient NFAs, meaning the synthetic cost of o-BTP-eC9 is the lowest among the commonly used NFAs.", "lower synthetic complexity" and Fig. 2d have been deleted.

We would like to point out that the o-BTP-eC9 end group (as shown in the below image) takes 2 steps to synthesis with a yield of 54%, in contrast, BTP-eC9 end group takes 3 steps with a yield of 36%. The difference does exist and should be meaningful for advancing the field. Using BTP-eC9-CHO, the center core, as the starting reactant, the yield difference is even obvious. Nevertheless, we agree that the center core BTP-eC9-CHO takes more steps, and the overall yield is low, we have minimized the discussions on cost.

Reviewer #3 (Remarks to the Author):

The paper reports the synthesis of a new non-fullerene acceptor, for use in organic photovoltaic devices. The design is a very small tweak on the existing NFA materials, in that the terminal end-group has a different isomer pattern of its two chlorine groups. There has been significant work on the nature of end-group isomers, and indeed this exact same molecule has already been published <https://onlinelibrary.wiley.com/doi/full/10.1002/advs.202207678> (May 12th) by one of the authors of this study (Shirong Lu). The only difference is the recent paper used a C11 sidechain on the central core, whereas this paper uses a C9.

The synthesis of the end group and its simplification compared to other dechlorinated end-groups are discussed in this paper (in a very similar manner to the current manuscript), and the dipole moment is also calculated and presented.

A synthetic complexity analysis is also presented in the published paper, as in the current manuscript. Interestingly, the same materials get a different SC score in each paper. For example, BTP-eC9 is 92.6 in the Wiley paper, but (table SS1) 79 here. Similar differences occur for other materials in each manuscript. Given that these are based on published analysis, the reason is not clear.

Response:

We thank the reviewer for the comment.

The Wiley paper mainly focused on the efficiency-cost gap, while the key points of this paper are the rational design and the in-depth device mechanism, which the Wiley paper is superficial. Besides, on device engineering, this work is much deeper and achieved much higher PCE (19.9% vs 17.9%) via rational molecular end group design and selection.

Although the change in our new molecule is also halogen substitution position, the highlights and contributions of this manuscript are very different from those of the Wiley paper. First, ternary strategy is the most widely used method to further realize the potential of binary benchmark OSC, however, the rational design of guest material for ternary is rarely studied. In this work, guided by theoretical calculations, we designed a suitable guest component from the end group calculation and achieved significant performance enhancement. Besides, we deeply studied the structure-property relationship and established the link between molecular structure and device performance. More importantly, the working mechanism behind the enhanced performance was systematically investigated, such as how the guest component optimizes the nano morphology of the ternary blend, and why the new molecule can suppress radiative and non-radiative recombination loss, a critical issue preventing OSC performance improvements.

As for the SC score, Riccardo Po et al. (*Macromolecules*, 2015, 48(3): 453-461) proposed the concept of the SC to partially quantify material costs, which has received widespread attention and recognition, such as: *Advanced Energy Materials*, 2020,10(43):2001864; *Joule*, 2021, 5(5):1209-1230; *Nature Communications*, 2021, 12(1): 5093; *Angewandte Chemie International Edition*, 2021, 60(23): 12964-12970; *Advanced Functional Materials*, 2022, 32(13): 2110159; *Science China Chemistry*, 2023, 66(4): 1101-1110; *et al.*). The SC is calculated by the following equation

$$SC = 35 \frac{NSS}{NSS_{max}} + 25 \frac{\log(RY)}{\log(RY_{max})} + 15 \frac{NUO}{NUO_{max}} + 15 \frac{NCC}{NCC_{max}} + 10 \frac{NHC}{NHC_{max}}$$

assessed through five parameters: (1) the number of synthetic steps (NSS), (2) the reciprocal yields of the monomers (RY), (3) the number of unit operations required for the isolation/purification of the monomers (NUO), (4) the number of column chromatographic purifications required by the monomers (NCC), and (5) the number of hazardous chemicals used for their preparation (NHC).

The NSS_{max} , RY_{max} , NUO_{max} , NCC_{max} , NHC_{max} , correspond to the maximum values of the studied system. For different systems, these maximum parameters are different, thus, the SC score of the same material may also change.

In the Wiley paper, $NSS_{max} = 17$; $RY_{max} = 0.0707$; $NUO_{max} = 26$; $NCC_{max} = 10$; $NHC_{max} = 31$.

In our work, $NSS_{max} = 22$; $RY_{max} = 0.051$; $NUO_{max} = 41$; $NCC_{max} = 14$; $NHC_{max} = 34$. So, the calculated SC scores are different.

Furthermore, for the synthetic complexity comparison of BTP-eC9, versus the current NFA; I find the different values surprising given the only change is the end group. Table SS1 reports ref 11 for the synthesis of BTP-eC9, although no details are given so one must go back to https://onlinelibrary.wiley.com/doi/epdf/10.1002/adma.201703080?saml_referrer for the synthesis of the end group. This only involves on column chromatography step as far as I can see. The reported synthesis of the new isomer involves 2 CC steps (as

reported by the authors). However, table SS1, gives both materials as only 6 CC. Since the core is the same, there must be an error somewhere. Please check.

Response:

We thank the reviewer for pointing out this mistake.

The original paper on BTP-eC9 reports the synthesis of the end group, in which the intermediate product was used without purification, that is why only one column chromatography is reported. The synthesis of the new acceptor here involves 2 column chromatography steps. We have corrected the mistake in the revised manuscript. BTP-eC9 takes 5 column chromatography steps, and o-BTP-eC9 takes 6 column chromatography steps.

More fundamentally, the number from this analysis must only be used very broadly, and quoting to two or even decimal place seems meaningless to me. The analysis is only based on the small-scale research synthesis, which is unlikely to be used for scale-up.

Response:

We thank the reviewer for the comment.

The SC values have been modified to integer form and the yield is adjusted with two significant digits in the revised manuscript. And combined with Reviewer 2's comment, we have removed the discussions about SC.

For the synthesis of the new end group, only ¹H and ¹³C is provided to prove structure. Please also include other relevant characterization to prove structure, such as mass spectra, melting point.

Response:

We thank the reviewer for the comment.

We have conducted more characterizations to further study the properties of θ -o-IC-2Cl group. **Fig. R2** (updated as **Fig. S119** in the Supplementary Information) is the mass spectroscopy result of θ -o-IC-2Cl, which is consistent with the chemical structure of the new end group.

Fig. R3 is the differential scanning calorimetry (DSC) thermogram of θ -o-IC-2Cl. As the endothermic peak is at 193°C, the melting point of the new end group should be around 193°C.

Fig.R2 Gas chromatography-mass spectrometer (GCMS) spectrum of θ -o-IC-2Cl.

Fig. R3 The DSC thermogram of θ -o-IC-2Cl with a heating rate of $10^{\circ}\text{C min}^{-1}$.

The authors comment in comparing to BTP-eC9, that ‘due to the orientation effect of 2,3-dichlorobenzoyl chloride, o-BTP-eC9 needs fewer synthetic steps, and has much higher reciprocity yield (5.1% versus 1.8%) than BTP-eC9 (Table SS1 in Supplementary Information 2).] This makes little sense to me – the only difference is the end-group. The authors prepare their end group in two steps, with an overall yield of 54%. The alternative diCl end group is prepared in three steps in 36% yield (with only one chromatography step). This is a better comparison. The total synthesis of the NFA is the same from that point one, and any differences in yield likely relate to user experience.

Response:

We thank the reviewer for the comment. The total yield of the product is related to the synthesis step and the yield of the final step.

To clarify this point, this sentence has been changed to “due to the orientation effect of 2,3-dichlorobenzoyl chloride, the synthetic procedure for θ -o-IC-2Cl is one step less

than that for β -o-IC-2Cl, and the over yield of the new end group (54%) is obviously higher than that of its counterpart (36%). Besides the condensation reaction of θ -o-IC-2Cl with dialdehyde gives a yield of 96%, which is also significantly higher than that of the β -o-IC-2Cl (53%).” in the revised manuscript.

Please report the optical absorption coefficient, both in solution and in thin-film for the new acceptor. This will help to further prove the increase in dielectric constant reported for the new acceptor.

Response:

We thank the reviewer for the insightful comment.

As shown in **Fig. R4** (updated as **Supplementary Fig. 13 c** and **13d** in Supplementary Information), the maximum absorption coefficients of o-BTP-eC9 solution and film are $2.49 \times 10^5 \text{ M}^{-1} \text{ cm}^{-1}$, and $1.77 \times 10^5 \text{ cm}^{-1}$, respectively.

Fig. R4 Absorption coefficient spectra of BTP-eC9 and o-BTP-eC9 in CF (a) and neat films (b).

Fig 3 – specify what FOM is being used.

Response:

We thank the reviewer for the comment.

FOM used in **Fig. 3** is the abbreviation of figure-of-merit, where $\text{FOM} = \text{PCE}/\text{SC}_{\text{D:A}}$, $\text{SC}_{\text{D:A}}$ is the synthetic complexity of donor and acceptor materials. But combined with Reviewer 2’s comment, we have removed the discussions about SC and FOM. So, in the revised manuscript, we have replaced FOM with year in the y-axis title of **Fig. 3d**.

Overall, clearly the selling point of the paper is the high efficiency of the blends, approaching record values, but the novelty in terms of molecular design is low, not least in light of the Adv. Sci paper the authors have recently submitted, which has significant overlaps with the current manuscript (and some contradictions in terms of SC analysis). Therefore, I think the current manuscript needs to be updated to address and

acknowledge the Adv. Sci paper, and the SC needs to be carefully checked. After these changes, the paper may be suitable for this journal.

Response:

We thank the reviewer for the comment.

When we first submitted this manuscript (Mar 2023, to Nature Energy and then transferred to Nature Communications), the Adv. Sci paper had not been published, so we did not acknowledge it. In the revised manuscript, this paper has been cited as reference 19.

As for the novelty, we would like to indicate this work focused on the design of guest components for the benchmark binary blend to achieve performance breakthroughs, the working mechanism behind the high performance, and the relationship between molecular structure and device performance. While the Adv. Sci paper paid more attention to the balance between efficiency and synthetic accessibility.

Regarding the SC analysis, as we explained in your first comment, the SC value of a material would vary with the capacity of the database. Because this part is not the highlight of our manuscript, combined with Reviewer 2's comment, we have removed the discussions on SC and updated this manuscript correspondingly.

REVIEWER COMMENTS

Reviewer #1 (Remarks to the Author):

The authors addressed most of the questions raised by reviewers. I recommend it for publication without further delay.

Reviewer #2 (Remarks to the Author):

The authors have addressed the concerns from the original version. The revised version is now suitable for publication.

Reviewer #3 (Remarks to the Author):

Significant changes have been made to the manuscript, and the removal of the section on synthetic complexity is an improvement. The authors arguments about the interpretation of SC are hard to understand - they are the authors of both papers, and yet they produce a different number for SC for the same material in both papers. Nevertheless, as this section is removed, no need for further discussion.

There are still some issues to resolve in my opinion. Firstly, as asked last time they should provide evidence to confirm the structure of oIC-2Cl versus alpha-o-IC-2C (i.e. the isomer where the malonitrile is on the other carbonyl). This is a crucial piece of evidence currently missing. They have adding mass spectra and DSC, but this only proves the malonitrile is attached - not where it is attached. My modelling of both isomers suggest both are planar, without any steric issues. Therefore it is not obvious why the ortho isomer will form only. The other isomer has an EWG ortho to the ketone, which can activate it. Please therefore convince that the correct structure is formed. For example single crystal XRD of this isomer.

Whilst this is the most important issue, some minor issues are:

In the synthesis of oIC-2CL no info is given for the chromatography solvent. Please provide. Does the reaction only give one isomer? Or are both prepared.

The authors add a large amount of DIB versus the active materials. Given it is a solid at RT, what happens to this material? Does it stay in the film?

In device section, please add specific ratio of ternary components used. Is the chloroform heated to dissolve components? How long, what temp. etc.

The processing solvent and concentration of PFNBr used is not specified.

(The changes are highlighted in red color in the revised manuscript. The comments by the editor and reviewers are in black color and our responses are in blue color.

Reviewer #1 (Remarks to the Author):

The authors addressed most of the questions raised by reviewers. I recommend it for publication without further delay.

Response:

We thank the reviewer for the positive comment on our work.

Reviewer #2 (Remarks to the Author):

The authors have addressed the concerns from the original version. The revised version is now suitable for publication.

Response:

We thank the reviewer for the positive comment on our work.

Reviewer #3 (Remarks to the Author):

Significant changes have been made to the manuscript, and the removal of the section on synthetic complexity is an improvement. The authors' arguments about the interpretation of SC are hard to understand - they are the authors of both papers, and yet they produce a different number for SC for the same material in both papers. Nevertheless, as this section is removed, no need for further discussion.

There are still some issues to resolve in my opinion. Firstly, as asked last time they should provide evidence to confirm the structure of o-IC-2Cl versus alpha-o-IC-2C (i.e. the isomer where the malononitrile is on the other carbonyl). This is a crucial piece of evidence currently missing. They have added mass spectra and DSC, but this only proves the malononitrile is attached - not where it is attached. My modeling of both isomers suggests both are planar, without any steric issues. Therefore it is not obvious why the ortho isomer will form only. The other isomer has an EWG ortho to the ketone, which can activate it. Please therefore convince that the correct structure is formed. For example single crystal XRD of this isomer.

Response:

We thank the reviewer for the comments.

To further confirm the structure of o-IC-2Cl, we have cultivated the single crystal of this end group. The CCDC number is 2312645 and the related files have been uploaded. **Fig. R1** (updated as **Supplementary Fig. 12** in the Supplementary Information) is the chemical structure of o-IC-2Cl from single crystal analysis. As we can see, the chlorine substitutions and the carbonyl group are on the same side, which is consistent with the chemical structure in the manuscript.

Fig. R1 The chemical structure of o-IC-2Cl from single crystal analysis (CCDC number: 2312645).

Whilst this is the most important issue, some minor issues are:

In the synthesis of o-IC-2CL, no info is given for the chromatography solvent. Please provide. Does the reaction only give one isomer? Or are both prepared.

Response:

We thank the reviewer for the comment.

The chromatography solvent is dichloromethane, which has been added in the Supplementary Methods part. Regarding the products, both two isomers were formed, but the o-IC-2Cl product was obtained in a high yield of 77% (purified), the other isomer was not collected in this reaction.

The authors add a large amount of DIB versus the active materials. Given it is a solid at RT, what happens to this material? Does it stay in the film?

Response:

We thank the reviewer for the comment.

Using DIB as an additive to improve non-fullerene OSC performance was first reported by our group (Nano Energy 84 (2021): 105862) and has been adopted in many efficient OSC systems by other groups (Nature Materials 21.6 (2022): 656-663, Advanced Science 9.14 (2022): 2200578, Nature Communications 14.1 (2023): 4148, etc.). This additive can interact with non-fullerene acceptors and form eutectic phase behavior, thereby increasing the crystallinity of the acceptors and enhancing device

performance. During the spin coating process, DIB would stay in the active layer after the evaporation of chloroform, as shown in **Fig. R2a** and **2d**. However, this additive would be removed via thermal annealing (TA) because of its good volatility, as shown in **Fig. R2b** to **2e**, with the evidence from us and other groups.

Fig. R2 The good volatility of DIB. Images of PM6:Y6 blend film with DIB after spin coating **(a)** and TA **(b)**. **(c)** Image of PM6:Y6 blend film w/o DIB after spin coating. The images were captured by a microscope CCD camera and illustrate the removal of DIB after TA. **(d)** FTIR spectra of DIB, PM6:Y6, PM6:Y6:DIB and PM6:Y6:DIB-TA. In the TA-treated PM6:Y6:DIB sample, DIB signals at 1460, 1066, and 990 cm^{-1} disappear, again verifying the removal of DIB. **(e)** The images of DIB film coated on the silicon wafer before and after TA. **Fig. R2a** to **2d** are the results from our previous work (Nano Energy 84 (2021): 105862) and **Fig. R2 e** is from Advanced Science 9.9 (2022): 2105347.

In the device section, please add specific ratio of ternary components used. Is the chloroform heated to dissolve components? How long, what temp. etc. The processing solvent and concentration of PFNBr used are not specified.

Response:

We thank the reviewer for the comment.

The weight ratio of the optimized ternary device is D: A₁: A₂ = 1: 1.05: 0.15, where D is the polymer donor PM6, A₁ is the host acceptor BTP-eC9, A₂ is the guest component o-BTP-eC9.

The precursor solution would be heated to 60 °C for an hour to fully dissolve the solutes then cooled to room temperature before use.

Regarding the PFN-Br solution, the concentration is 0.5 mg/mL with methanol as the solvent.

Correspondingly, in the revised manuscript, the device section has been updated with these experimental details.

REVIEWERS' COMMENTS

Reviewer #3 (Remarks to the Author):

The additional data is convincing for the structure proof and all other questions has been clearly addressed. I recommend acceptance.